# Understanding health system responsiveness to public feedback at the sub-national level: Insights from Kilifi County, Kenya

Nancy Kagwanja[1]*, Lucy Gilson[2,3], Benjamin Tsofa[1], Jill Olivier[2], Hassan Leli[4], Sassy Molyneux[1,5]

1 Health Systems and Research Ethics Department, KEMRI-Wellcome Trust Research Programme, Kilifi, Kenya, 2 Health Policy and Systems Division, School of Public Health, University of Cape Town, Western Cape, South Africa, 3 Department of Global Health and Development, London School of Hygiene and Tropical Medicine, London, United Kingdom, 4 County Department of Health, Kilifi County Government, Kilifi, Kenya, 5 Centre for Tropical Medicine and Global Health, Nuffield Department of Medicine, Oxford University, Oxford, United Kingdom

* Nkagwanja@kemri-wellcome.org

**Data Availability Statement:** We are unable to make the data more freely available because the qualitative data reported in this article contains potentially identifiable information for individuals,

## Abstract

Responsiveness is one of four health system goals alongside health outcomes, equity in financing and efficiency. Many studies examining responsiveness report a composite satisfaction index or proportions of patients describing satisfaction with dimensions of responsiveness. Consequently, responsiveness is predominantly based on collation of service users' feedback and could be termed *service responsiveness*. We conceptualise *system responsiveness* more broadly, as how the health system more widely responds to concerns or needs of the public. In this paper we share a system responsiveness framework to reflect this wider conceptualisation and illustrate how we used this framework combined with Aragon's insights on organisational capacity, to explore system responsiveness practices at sub-national level in Kenya. Drawing on interviews and group discussions we specifically consider how two governance structures -Health Facility Committees (HFCs) and Sub-County Health Management Teams (SCHMTs)- found in many Low-and-Middle-Income (LMIC) health systems receive, process, and respond to public feedback. HFCs are formal structures with community representation linked to a health facility to support community participation in service provision and health outcomes. SCHMTs comprise middle-level managers with oversight over primary health care facilities and are commonly known as district health management teams in other LMICs. There were multiple feedback mechanisms through which the health system could receive public feedback, but these mechanisms had limited functionality, often worked in isolation, and inadequately represented vulnerable groups. Our analysis also revealed the organisational capacity gaps that constrain health system responsiveness. These gaps ranged from inadequate funding and staffing of feedback mechanisms (hardware), through absence of clear procedures and guidelines (tangible software), to norms, actor relationships and power dynamics (intangible software elements). Our findings are relevant to similar low-and-middle-income contexts and draw attention to the importance of integrating multiple mechanisms and forms of feedback,

health facilities, and organisations. The ERC-approved protocol and participant consent forms did not include obtaining explicit consent regarding the possible use of anonymised data in the public domain via a data repository. Questions about data access can be directed through the Data Governance Committee of the KEMRI/Wellcome Trust Research Programme at DataGovernancecommittee@kemri-wellcome.org.

**Funding:** The research in this article was nested in a broader study funded by the United Kingdom Medical Research Council 'Strengthening health system responsiveness to community and citizen feedback in SA and Kenya' (UKRI: MR/R013365/1). This research was also funded in part, by the Wellcome Trust [DEL-15-003] and the UK Foreign, Commonwealth & Development Office, with support from the Developing Excellence in Leadership, Training and Science in Africa (DELTAS Africa) programme. At the time of doing this study, NK was a recipient of a Health Policy Analysis Fellowship by the Alliance for Health Policy and Systems Research which provided a bursary award towards this research. The funders had no role in the decision to publish this work.

**Competing interests:** The authors have declared that no competing interests exist.

alongside considering system capacities and their interactions, in strengthening health system responsiveness.

## Introduction

Responsiveness is one of four health system goals alongside health outcomes, equity in financing and efficiency introduced by the World Health Organisation (WHO) in the World Health Report 2000 [1]. In this framework, health *system* responsiveness is largely understood as linked to the interaction between individual users or patients and health *services* [2]. It can contribute to health and well-being by providing an environment in which the public seek care early, interact positively with healthcare providers and incorporate health information into their lives [3]. Studies that have drawn on this framing of responsiveness have generally adopted an evaluative approach, utilising surveys to collect feedback from patients after they have used services [4–7]. These studies commonly report on a composite satisfaction index or on proportions of patients describing satisfaction with dimensions of responsiveness [4–7]. As a result, responsiveness is predominantly based on collated individual-level feedback from service users and could be termed 'health service responsiveness'.

In this study, we conceptualise *responsiveness* as how the health system more widely reacts/responds to the needs and concerns of the public [8]. Such responsiveness can build trust in the health system and contribute to a more inclusive and accountable health systems [9–12]. Beyond these instrumental benefits, responsiveness also has intrinsic value [1], which is consistent with an understanding of health systems as people-centred and a social good [12, 13]. We adopt a system lens in line with recent calls [2, 14]. In so doing, we go beyond service delivery encounters and satisfaction levels to pay attention both to actors and processes and to system hardware and software. Health system software includes tangible (skills, knowledge, decision-making processes), and intangible (values, norms, relationships, and communication practices) elements that are distinct from but interact with hardware (funding, staffing, technology) [15–17]. Our approach, explained in more detail in the methods section, also includes exploration of how Health Facility Committees (HFCs) and Sub-County Health Management Teams (SCHMTs) in Kenya [18], two important governance structures, receive, process, and respond to public feedback, and what interactions occurred between them across the *responsiveness pathway*. We understand that such a pathway includes receiving, processing (for example, through analysis, integration, and/or prioritization) and responding to feedback [8]. The term *feedback* refers to the input, views and concerns raised by the public, while a *feedback channel* is the mechanism through which these views, concerns and input reach the health system.

We selected HFCs for in-depth exploration of their roles related to responsiveness because, in many low- and middle-Income countries (LMICs), they are common mechanisms introduced to support community participation [11]. Many HFCs are linked to Primary Health Care (PHC) facilities and are elected by communities to act as a link between the facility and community [11, 19]. HFC roles include support for public voice and 'integration of the public's preferences in health system decision-making' [20], but these roles have received relatively little attention in literature. Similarly, while several studies have focused on decision-making and management experiences of sub-national Health Management Teams [18, 21–23], such as SCHMTs, few have considered how these teams receive and respond to public feedback [24].

Overall, we aimed to answer the research questions: How is public feedback received, processed, and responded to, and what influences the practice of responsiveness at sub-national level?

## Materials and methods

### Study setting

This study was conducted in Kilifi County, Kenya. Kenya has a devolved government system comprising a national government, and 47 semi-autonomous devolved county governments [25]. Within the health sector, the national MoH is responsible for health policy formulation, training and regulation of health services while county governments have responsibility for service delivery [25]. County Health Management Teams (CHMTs) and SCHMTs provide oversight, manage and plan service delivery at county and sub-county levels respectively [26]. Fig 1 below developed from work on accountability relationships done in Kilifi County highlights the connections and reporting arrangements between various institutions that interact with and within the county health system [26].

The decision to use one county was to allow for a deeper exploration of the issues under focus within the study. Kilifi county has an estimated population of 1.5 million people [27], and high rates of poverty and inequality [28]. Kilifi County was a good fit for this study because of the embeddedness of health policy and systems researchers (BT and NK) within the county health system [29, 30]. This embeddedness supports selection of relevant topics and a nuanced understanding of the context and can support translation and utilisation of research [31]. Further, the long-standing relationship has enabled trust-building which was key for gaining access to study participants and documents for review, and for building the rapport required to conduct this qualitative study, which sometimes involved discussion of sensitive topics [30].

Table 1 below presents the key demographic and health indicators of the county.

### Conceptual framework

Drawing on a literature review of health system responsiveness [2], we conceptualised responsiveness as comprised of three interrelated processes—receiving, processing, and responding to public feedback–and these occur in, 'processing spaces' within the health system (Fig 2). Fig 2 highlights that feedback may come from various public groupings, including marginalized groups (circle 1). Considering who feedback comes from allows for the examination of inequities in responsiveness [1, 32]. Receiving feedback can occur through engagements between the

**Table 1. Kilifi county health and demographic indicators.**

| Indicator | Kilifi county 2018 |
| --- | --- |
| **Population** | |
| Total | 1, 498, 647 |
| Male | 723, 204 |
| Female | 775, 443 |
| Under 5 | 54, 518 |
| Under 1 | 259,538 |
| **Healthcare workers** | |
| Nurses (per 10,000 people) | 4 |
| Doctors (per 10,000 people) | 1 |
| **Health Facilities** | |
| Public | 150 |
| Faith-based | 13 |
| Private | 135 |

Source: Kilifi County Integrated Development Plan (2023–2027)

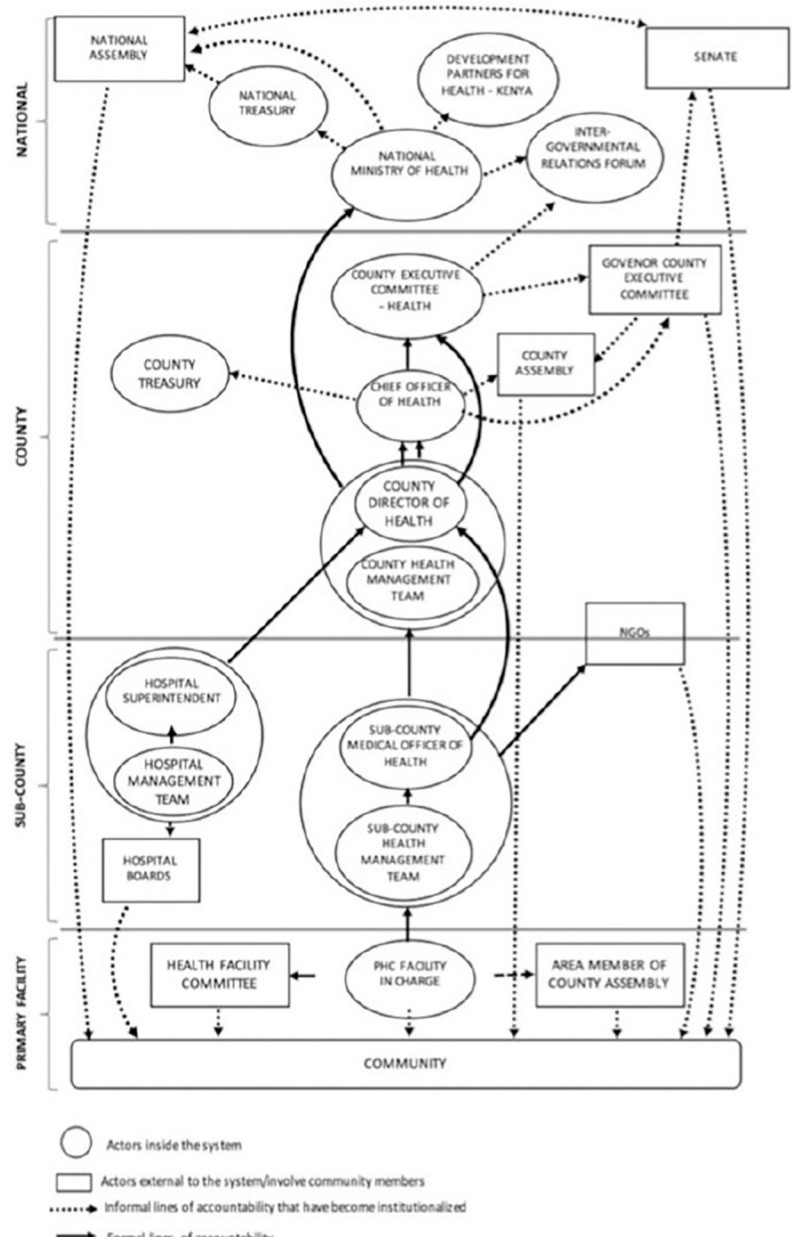

**Fig 1. Management and governance structure and reporting responsibilities within the county health system including national level (adopted from Nxumalo et al, 2018) [26].**

public and health system actors across varied channels (both formal e.g. the participatory, and unidirectional channels and informal). Formal mechanisms are those outlined in policy documents, while informal mechanisms are additional ones that arise in practice, sometimes due to absent or weak functioning of formal feedback mechanisms [33, 34]. Diverse responses may be enacted by policymakers, system managers and/or service providers (Fig 1, circle 2), and these could range from information or action, changes in the mechanisms of processing feedback, to inaction (Fig 2, circle 3). Linked to the idea of system change, we consider health *system* responsiveness as requiring a response at the system level and not just at the point of interaction between individual service users and providers [14, 35]. Such responses might include for

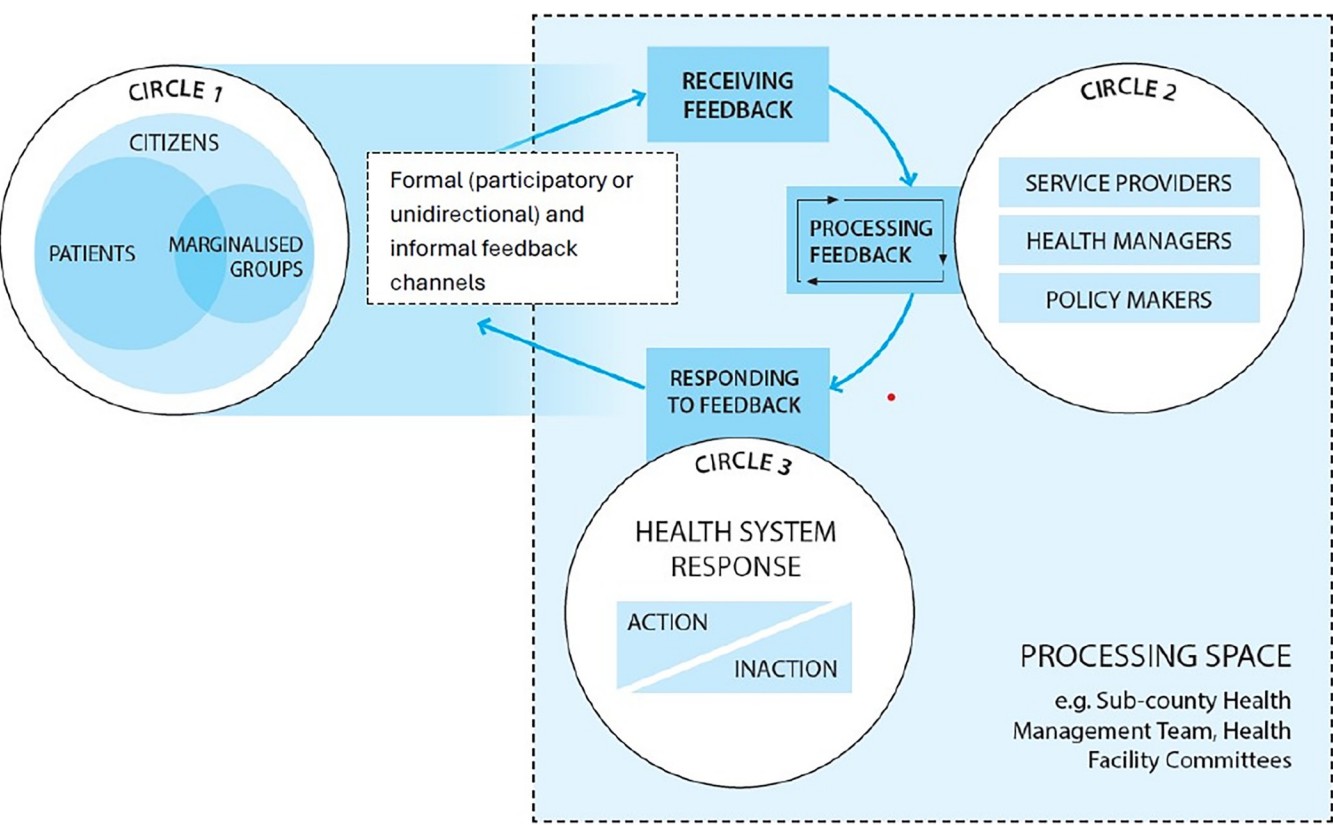

**Fig 2. Conceptual framework illustrating processes of receiving and responding to public feedback.**

example a change within a department in a facility, or at facility-level, or across multiple facilities in a sub-county.

## Qualitative case-study approach

We adopted a case study approach because of its suitability for in-depth and holistic exploration of a complex issue [36, 37]. The "processing spaces" illustrated in Fig 2 (boundary marked by dashed lines) served as the "cases" of focus in this study–that is, Health Facility Committees (HFCs) and Sub-County Health Management Teams (SCHMTs). HFCs are comprised of community members, health managers, and political and administrative representatives, while SCHMTs are composed of health managers. HFCs work at primary healthcare facility-level while SCHMTs co-ordinate service delivery across multiple PHC facilities in one sub-county. More descriptive details about the case study SCHMT and HFC and their linked sub-counties and facilities are provided in S1 File. These governance structures were purposively selected to support examination of system-level interactions as, in the study context, SCHMTs have oversight responsibilities for HFCs [38, 39]. We selected two HFCs per SCHMT (Fig 3) to allow for in-depth exploration within the available time and resources. To protect confidentiality, the SCHMTs and HFCs considered are identified with numbers and letters (Fig 3), with the Primary Health Care facilities linked to the case study HFCs identified as Facility 1A, 1B, 2A and 2B.

We conducted 35 in-depth interviews and four focus group discussions with a range of respondents (sub-county health managers, facility in-charges and frontline providers, and Members of the County Assembly (MCAs) and HFC members (Table 2). MCAs are local political representatives who have legislation and oversight responsibilities within the County [40].

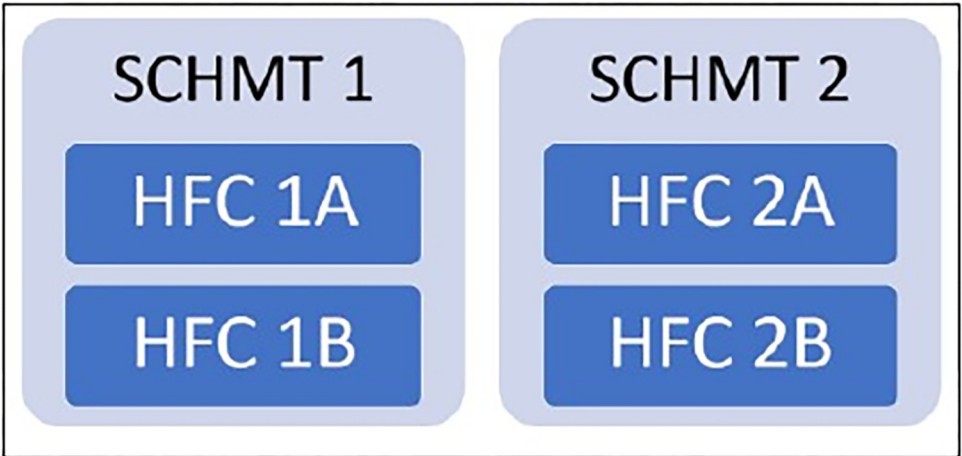

**Fig 3. Cases for in-depth exploration.**

They also serve as ex-officio HFC members. The respondents interviewed were purposively selected for their involvement in receiving, processing, and responding to citizen feedback across all the case study SCHMTs and HFCs. The interview and FGD topic guides included questions about the nature of feedback received by the HFCs and SCHMTs, what channels members of the public utilised to provide feedback, whether and what responses were generated to this feedback (S1 File). Other forms of data collection included observations of SCHMT activities and a review of SCHMT and HFC minutes as summarised in Table 2. The observation guide included prompts to observe for the SCHMT meeting setting, interactions between SCHMT members, the content of their discussions during meetings and support supervision visits, and the frequency of their meetings and activities (S2 File).

## Data analysis

After transcribing interviews and focus group discussions, data were imported into NVIVO12 to support analysis. Data analysis was led by the first author, with support from all authors. We

**Table 2. Summary of data collected.**

| Data collection activity | Quantity/ Duration Respondents |
|---|---|
| In-depth interviews | Sub-county health management team members (16)- |
| | County health Management team members (2) |
| | Health facility managers (5) and frontline workers (8) |
| | Members of County Assembly (4)-3 linked to health facilities 1A, 1B, & 2A |
| | 1 member of the Health Services Committee |
| Focus Group Discussions | Four with Health facility committee members (except in-charge) |
| Observations of meetings | Observation of Sub-county health management team meetings & support supervision (SCHMT-1) between July and August 2021 (6 meetings) |
| Document review | • County level documents (Count Budget Outlook Paper, County Integrated Development Plan; CDIP, Health Sector Mid-term Review, County Budgets) |
| | • Sub-county health management and health facility committee minutes |
| | • Sub-county team & Health Facility Annual Work Plans |

adopted a framework approach to data analysis because it supported the systematic treatment of similar units, and enabled comparison between and within cases [41]. To understand the practice of responsiveness, we coded for data related to feedback mechanisms, their functionality, and processing of and responses to feedback. In charting the data into various categories, we drew on the distinction made by Ortiz Aragon and adapted by other health policy and system researchers between interacting components of organisational capacity: the hardware of technology, infrastructure and funding; the tangible software of knowledge, skills and processes of decision making; and the intangible software of relationships, communication practices, power, values and norms [15–17]. System hardware are visible and quantifiable components [17] and are more frequently reported on, while system software [both tangible and intangible] are less frequently reported on yet are critical to the performance of health systems. In this study we consider both, as well as the interaction between these components to highlight the complexity of responsiveness and identify challenges to and opportunities for strengthening health system responsiveness.

### Ethical considerations

This study obtained ethics review and approval from the KEMRI scientific and ethics review committee, [SERU/CGMR-C/171/3920]. Permission to conduct the research was also obtained from the County Department of Health. All participants provided written informed consent to participate in the study.

### Inclusivity in global research

Additional information regarding the ethical, cultural, and scientific considerations specific to inclusivity in global research is included in the S1 Checklist.

## Results

Our study findings are presented in three broad sections in line with the elements of responsiveness outlined above. These findings demonstrate the multiplicity but inadequate functionality of feedback mechanisms and highlight the influence of system hardware and software on responsiveness to public feedback. We also present the content of public feedback received and consider to what extent the experiences of vulnerable groups were included.

### Receiving public feedback

**Many feedback mechanisms but weak functionality.**   There were multiple channels through which health system actors could receive public feedback. Box 1 below distinguishes

**Box 1. Feedback mechanisms available in Kilifi county**

| Participatory mechanisms within the health system• Public participation meetings held at county level• County health board• Hospital boards• Health facility committees in PHC facilities• Civil Society Organisations s working within sub-counties• Community health committees at community level<br>• Community strategy-Community Health Committees—Community Health Volunteers-Community Health Assistant-<br>• Subcounty Health Management Team and County Health Management Team | Uni-directional Service feedback mechanisms<br>• Sub-county complaints committee<br>• Annual client satisfaction surveys at health facility level<br>• Suggestion boxes/complaint boxes at facility level<br>• Hotlines at facility level<br>• Service charters at facility level |

between participatory and uni-directional mechanisms. In this work, we judged participatory mechanisms to be those in which the public are invited to contribute feedback, and there is opportunity for bi-directional or two-way engagement between the public and health system actors. Uni-directional mechanisms support the public to share feedback with health system actors but mainly involve collection of feedback, thus feedback flows in one direction and there is little engagement at the point of providing feedback. Box 1 includes county-level mechanisms because it was not uncommon for public feedback related to facility-level experiences to flow from higher health system levels to lower-level facilities.

Despite these multiple feedback mechanisms, study respondents perceived that little public feedback was received and incorporated into health system decision making. Table 3 below summarises functionality details about feedback mechanisms and highlights various system hardware and software constraints that cut across both the uni-directional and participatory mechanisms. In summary these included hardware issues related to funding and staffing while software issues related to the processes through which feedback was handled, for example whether there was documentation, how feedback and feedback channels were managed, and whether co-ordination and prioritisation of feedback occurred. These are explored in more detail in the sub-sections below. Quotes reflecting the themes discussed subsequently are presented in Table 4.

**Table 3. Summary of functionality challenges across feedback mechanisms.**

| Feedback mechanism | Hardware-related constraints | Software-related constraints |
|---|---|---|
| Unidirectional service feedback mechanisms | | |
| Suggestion boxes | • None of the suggestion boxes in the PHC facilities visited had a paper and pen in proximity for members of the public to use<br>• There was no dedicated staff with responsibility for opening the suggestion box | • Suggestion boxes were rarely opened. At the time of data collection between October and December 2021 suggestion boxes had not been opened in all four-case study HFCs in the two preceding quarters (Apr-Jun 2021 & Jul-September 2021) |
| Client satisfaction surveys | • No funding allocations to conduct client satisfaction surveys in PHC facilities. | • Neither SCHMT had records of previously conducted satisfaction survey findings. Documentation of findings was held by NGOs who initiated them. |
| Hotlines | • New hotline phone numbers introduced during the COVID-19 pandemic were deactivated due to non-payment to the service provider<br>• No dedicated staff to manage hotlines | • Weak co-ordination of feedback shared through hotlines to ensure response generation. |
| Sub-county complaints committee | • Lack of quorum among SCHMT members to set up a sub-committee for complaints and other public feedback | • Low awareness across both SCHMTs about membership and terms of reference for committee to handle public feedback<br>• No documentation of feedback received directly at SCHMT level |
| Participatory feedback mechanisms | | |
| Health Facility Committees (HFCs) | • Many newly elected HFC members had not received training on their roles due to funding constraints | • Selection processes of the HFCs resulted in weak representation of youth and Persons living with disability<br>• Mainly passive process of receiving feedback<br>• No documentation of public feedback<br>• Little awareness among HFC members of feedback received via other mechanisms |
| Community Strategy Structures -Comprising CHCs, CHVs, and CHAs | • Limited coverage of community units thus some areas had no or too few CHVs<br>• CHCs were mainly inactive or were not set-up within community units | |
| Public participation Forums | • Public participation for budgeting and planning was limited to one day for all ten county departments. | • SCHMT attendance of public participation meetings was inconsistent.<br>• Response to issues raised by the public was limited by low awareness among SCHMTs of overall health budget and choices made |

CHA-Community Health Assistant; CHC-Community Health Committee; CHV-Community Health Volunteer; HFC-Health Facility Committee; SCHMT-Sub-county Health Management Team

**Table 4. Factors influencing feedback mechanism functionality and their effects (quotes from participants, across cases and by theme).**

| Quote Number | Perspective | Themes | Quote |
|---|---|---|---|
| 1 | Health facility-in-charge (HFC-002) | Hardware barriers to feedback mechanism functioning (funding) | *"Yeah, that is ideal, after being elected, they should be trained on their roles, but now we have not been having funds for the same for the last 6 years. When health was devolved, we started lacking the funds to train them [HFC members]. So they just serve like that, but we do an orientation, the sub-county team calls a few of them, the chairperson and the treasurer. The other members don't go. The training should be 5 days, but they get a one-day orientation"* |
| 2 | Sub-county health manager (SCHMT2 | Hardware barriers to feedback mechanism functioning (funding) | *"I put up the request [for HFC training] it goes through the processes to the treasury then when it reaches the treasury there is no money. You wait for like over a year in fact that money [for training] may not come at all so that is the biggest challenge because any time money arrives at the treasury, they have other priorities like people the suppliers have not been paid" [SCHMT1-009]* |
| 3 | Sub-county health manager | Hardware barriers to feedback mechanism functioning (funding for training HFCs) | *"Many of our committees [HFC] are not very strong and it's because of the way the structure has been. Because when you've not given them their roles and responsibilities. . .if the health care workers behave a way that does not please the community, they [HFC] should come [to the SCHMT]. . .or even before coming to report they should sit with the healthcare workers at the facility telling them that we have observed this and this which we feel is not right but many times the health care worker becomes like their boss so they are at the mercies of the health care worker which is not right" (SCHMT1-009)* |
| 4 | Sub-county health manager | Hardware challenges undermined receiving public feedback | *"The staff providing services are often overwhelmed and they feel aggrieved because their welfare is not catered for. With two staff on duty at a health facility who are expected to run four departments (maternity, Child Welfare Clinic, OPD, HIV) there are bound to be complications when the staff divide the departments between themselves. If a complication arose in maternity, patients waiting to be served in all other departments will have to wait. If you do a satisfaction survey at this time you will not get the real picture of the facility." (SCHM2-004)* |
| 5 | Sub-county health manager | Weak tangible software (passive approach to receiving public feedback) | *"We are supposed to do client satisfaction interviews at dispensaries and health centers. It is one of the performance indicators that we need to track, the same way we track staff meetings, facility management committee meetings, we should also be doing that, but we rarely do that. I think maybe we have not given it the seriousness that it deserves. It is not seen as something very important. If the patient has a problem, they will state it, the problem is solved, and people carry on" (SCHMT1-001)* |
| 6 | Sub-county health manager | Software barriers to response generation: power dynamics | *"As the presenter [SCHMT member who had attended a public participation forum] I don't have all the powers to say fine, we will not open facility C, we will equip the level four facility within your ward for better service provision. I would have now to give that feedback to my supervisor, and the supervisor now forwards it to the CHMT for consideration" (SCHMTA-005).* |
| 7 | Sub-county health manager | Software barriers to response generation: Communication flow | *". . .we don't get feedback that this can be acted on, and this cannot, and why it cannot be acted, we need to get that feedback," (SCHMT1-01)* |
| 8 | Sub-county health manager | Interactions between hardware and software hindered response generation | *"I think that you cannot entirely blame the staff [for providing little information to the public] because of how the [health] system is. Because when you have a hundred or fifty patients waiting [and] you hardly have fifteen minutes it's difficult to give a lot of information. But again, I think there are those. . . I have interacted with some colleagues who say you'd rather tell them to come tomorrow and deal with five—that I will give real quality care. But how many of us will do that?" [SCHMT1-006]* |
| 9 | Sub-county health manager | Interactions between hardware and software hindered response generation | *"In school, we [health providers] are never taught in fact. . . and in some schools, that bit of communication is never there. The frustration starts in school. First, you know when there is a senior consultant around then for you there isn't much you can do [but watch how they do things]. So, there isn't I mean that kind of communication course for how to communicate to your clients. . .. Then we come here and now we feel like now the client is under our mercies. You know so your word is final, they have no opinion in their management, in their treatment, in their medical care and we believe now you own the client instead of giving them that space to participate in their treatment and medical processes. . ." [SCHMT1-006]* |

*(Continued)*

**Table 4.** (*Continued*)

| Quote Number | Perspective | Themes | Quote |
|---|---|---|---|
| 10 | Sub-county health manager | Weakly functioning formal feedback mechanism contributed to informal feedback mechanisms | '*Social media concerns are so many, and they usually occur at an unexpected time. First, you need to be online. . . but once we see them because they usually raise a lot of political pressure, they are usually handled very fast. An example, the other day, the Chief [Officer] and the head of Preventive [services] were at a health facility just because of a report on social media. The social media feedback of the community attracts a lot of political pressure. . .it is not good. . . because things usually are not the way they are reported [SCHMT1-003].* |
| 11 | Sub-county health manager | Emergence of informal feedback mechanisms | "*The member of the public calls the MCA who calls the governor, the governor calls the department [County Department of Health senior official], 'do this and this'. . . they impose what must be done, and you know there are others that need that service, and they are waiting in line, and this connected person calls so that they are served first. That's very unfair you see.*" (SCHMT2-003) |

## Hardware barriers constrained overall mechanism functioning, interacting with intangible software

From Table 3 above, hardware barriers to the functionality of these mechanisms included scarce financial resources to establish and sustain feedback mechanisms, and limited staffing to support their functioning. Funding barriers cut across both the unidirectional and participatory mechanisms and were expressed in relation to the conduct of satisfaction surveys, training of HFCs, and formalisation of CHCs and CHVs. For example, SCHMT members acknowledged that satisfaction surveys were rarely initiated by the County Department of health, because of the additional resources required to carry them out. HFCs also experienced funding constraints. In the early days of their establishment, HFCs received significant funding support from a development partner. However, HFC members elected over the last two election cycles were not comprehensively trained on their roles because SCHMTs experienced challenges in accessing funding (*Quotes 1 and 2*, Table 4). To cope with funding constraints, SCHMTs relied on implementation partners and donors to support initial establishment and functioning of feedback mechanisms. For several feedback mechanisms, functioning either stalled or slowed down significantly between grant periods or when partners moved out of the sub-county, linked to government failing to allocate funds to take over expenses.

Table 3 also highlights that many of the feedback mechanisms experienced staffing challenges, another hardware issue. Receiving public feedback was not a formally assigned role across most facility-level mechanisms. For example, inconsistencies in reports regarding who and when the suggestion boxes were opened, suggested that were rarely opened, and little feedback was received through them.

At sub-county level, several SCHMT members had multiple roles, raising questions about the teams' capacity to effectively pay attention to public feedback. For example, in SCHMT-1, the Sub-County Health Administrator served two sub-county teams and one sub-county hospital. Several members of SCHMT-1 also had coordination roles for programmes across the county (programme officers). In SCHMT-2, to cope with staffing challenges, SCHMT-2 programme officers were required to work from PHC facilities so that they could co-ordinate programmes at the sub-county level while offering services at the frontline. These multiple functions by individual SCHMT members could explain the lack of dedicated support for HFCs at SCHMT-level. This, together with funding constraints, resulted in HFCs that had not been trained and that were perceived by several SCHMT members to be weak, dominated by health providers, and with a low understanding of their roles (*Quote 3*, Table 4).

While the hardware barriers limiting functioning of feedback mechanisms were the more visible constraints to receiving public feedback, subtle influences from intangible software elements such as power appeared to underpin whether funding was available to support feedback mechanisms. For example, despite planning and budgeting for HFC training, it was not unusual for the more powerful County Department of Finance to prioritise other payments over release of training funds to SCHMTs (*Quote 2*, *Table 3*). Interactions between system hardware and software appeared to influence the extent to which public feedback was prioritised. For example, hardware problems related to funding and staffing appeared to shape SCHMT attitudes to public feedback and might have influenced why satisfaction surveys were hardly initiated by the County Department of Health. SCHMT-2 respondents perceived that any satisfaction survey conducted would only highlight the negative aspects of service provision due to long-standing health system challenges such as healthcare worker shortages, and de-motivated staff (*Quote 4*, *Table 4*). This suggests that public feedback was a low-priority issue given the existing staffing challenges and related health worker demotivation which were perceived to be far more pressing issues. According to several sub-county health managers, '*they [health managers] were already aware*' of most issues that would be picked up from the few satisfaction surveys that were conducted. These views combined with sentiments across both SCHMTs that conducting surveys would require additional resources from an already under-resourced health system appeared to keep the SCHMT from planning for the conduct of satisfaction surveys specifically and oriented them away from public feedback more generally.

### Receiving and processing of public feedback

**Weak tangible software underpinned the lack of systematic processing of public feedback.** Despite rhetoric that the SCHMTs and HFCs valued public feedback, they lacked a pro-active, consistent, and systematic approach to public feedback management. First, at HFC-level, most HFC members reported receiving feedback primarily from their friends, relatives and members of the public who knew them, suggesting that the voices of a significant segment of the population were excluded. It was also uncommon for both case study SCHMTs to actively seek out public feedback. For example, in explaining why client exit surveys were hardly conducted a sub-county health manager alluded to a passive approach to receiving public feedback (*Quote 5*, *Table 4*).

Second, the different feedback mechanisms that could have 'fed' the case study HFCs and SCHMTs appeared to function in silos. SCHMT respondents reported little linkage between facility-level HFCs and Community Health Committees (CHCs). Yet, CHCs were on paper expected to link with HFCs to support receiving feedback from a broader segment of the population. CHCs are the Community Strategy's governing structure and are comprised of 13 community members with oversight over implementation of community-level health service delivery by community health workers [42]. At HFC level, there was little awareness about public feedback picked up by other mechanisms such as suggestion boxes or satisfaction surveys.

Third, there was little documentation of feedback received by both HFCs and SCHMTs. Where HFC members conducted monitoring visits of facilities, these visits were informal, and findings were communicated to the facility-in-charge via word of mouth. The visits were also infrequent, save in the case of HFC-2B where the facility-in-charge reported weekly visits by HFC members. Across the case study SCHMTs and HFCs there were often no records available of receiving or responding to public feedback. Consequently, there was little analysis of feedback received to determine trends over time or across facilities. It was unclear too; how public feedback was prioritised for response.

A fourth barrier to processing public feedback was limited awareness and availability of guidelines and policies for handling public feedback. For example, CHMT respondents reported the existence of a county complaints policy and terms of reference for a sub-county complaints committee (charged with responsibility for receiving, deliberating on, and generating responses to complaints raised by the public). However, only one SCHMT-1 member reported awareness of the TOR and complaints policy. There were also no guidelines for how frequently the unidirectional feedback mechanisms, should be accessed to learn about public feedback. For the HFCs, there was limited material (guidelines or a manual) at facility-level that could be used to support ongoing familiarisation with their roles.

**Feedback received depicted negative experiences with the health system and limited views from vulnerable groups.** Despite the limitations described above, some public feedback was received but this was disproportionately negative and was commonly reported as though voiced by a homogenous public, making it difficult to identify specific feedback from vulnerable groups. Fig 4 below illustrates the content of public feedback as reported by SCHMT members, most of which mirrored HFC-level feedback. The feedback provided cut across four broad issues: healthcare worker conduct and performance (for example issues related to communication and with service users and their availability at the health facilities); service delivery processes (such as duration and availability of services); commodity and infrastructure related requests and resistance to uptake of services. Resistance to public health initiatives is included in Fig 4 below because it was judged to be a form of feedback that reflects experiences with engaging with the health system.

The experiences of some vulnerable groups were picked up mainly by channels within the health system dedicated to these groups, often supported by NGOs. For example, *mama* (mother) *open days* and youth forums picked up feedback related to the experiences of pregnant women and youth respectively. Notably, the experiences of other vulnerable groups such as People Living With Disability (PLWD) rarely featured in respondents' reports. Though PLWD were frequently mentioned by study respondents as a vulnerable group whose inclusion in HFC membership was a legal requirement, none of the case study HFCs had a PLWD

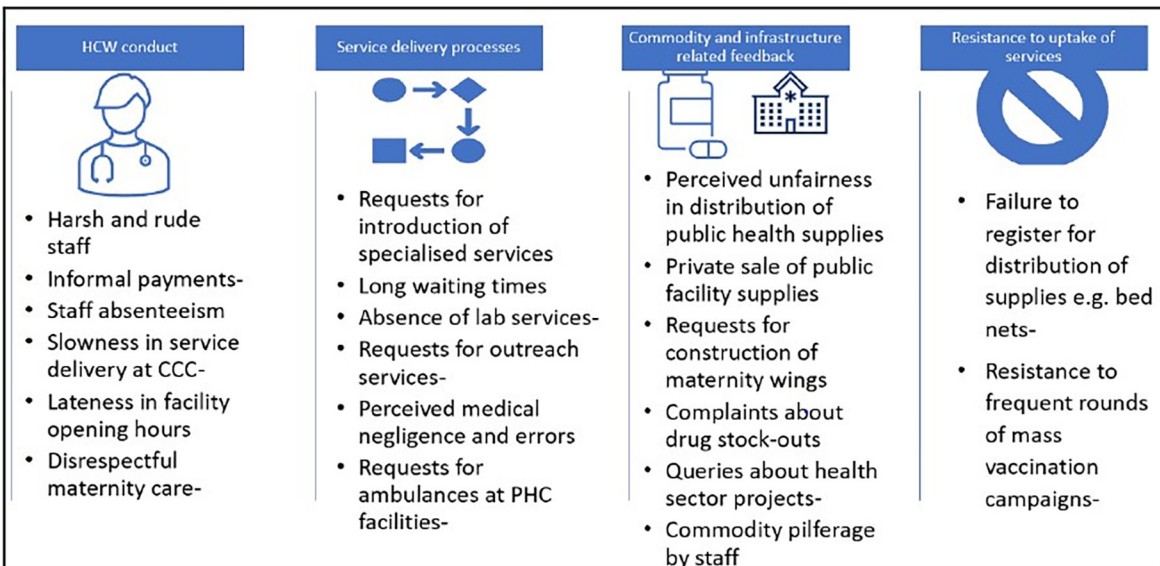

**Fig 4. Content of feedback received at SCHMT level (Source: Authors).**

**Table 5. HFC membership characteristics.**

| Characteristic | HFC1A | HFC1B | HFC2A | HFC2B |
|---|---|---|---|---|
| No. of elected members | 5 | 7 | 9 | 9 |
| No of active community members | | | | |
| Male | 2 | 5 | 7 | 7 |
| Female | 1 | 2 | 2 | 2 |
| Youth representative | 1 | 1 | 0 | 0 |
| PLWD representative | 0 | 0 | 0 | 0 |
| Exofficio members: | Members of County Assembly | | | |
| | Ward Administrator | | | |
| | Chief and/or assistant chief | | | |

Source: Document review (HFC minutes), in-depth interview and FGD data

Abbreviations: PLWD-People Living With Disability

representative (Table 5). The absence of PLWD in the committees may have contributed to the observed limited feedback from them.

Other vulnerable groups, for example, minority ethnic communities were also not included in HFC membership. Several HFC and SCHMT members perceived that these groups could share feedback through mechanisms specific to them such as Disabled Peoples' Organisations for PLWD and ethnic spokespeople for the minority communities. However, it was unclear how well these mechanisms (typically external to the health system) fed back into the health system.

## Responding to public feedback

**Varied, incident-driven responses to formal and informal public feedback.** Table 6 below presents a range of responses that varied with the form and perceived magnitude of public feedback. Moving from left to right, Table 6 illustrates responses generated at the PHC

**Table 6. Range of responses generated across case study HFCs and SCHMTs.**

| Form/type of feedback and respective response | Variation of responses | | |
|---|---|---|---|
| | **Responses enacted by HFCs** | **Responses enacted by SCHMTs** | **Escalation to higher level authorities by both HFCs and SCHMTs** |
| Healthcare worker conduct | • Dialogue and Mediation delegated to facility-in-charge by HFC members to respond to poor HCW conduct | • Dialogue and mediation <br> • Transfers across PHC facilities for persistent negative HCW conduct | • Reports to CHMT for perceived serious medical errors by HCWs (negligence) |
| Service delivery | • Modifications to improve service delivery-e.g. instituting sign-in books to address delays in service delivery <br> • Introduction with facility-in-charge of specific clinic days for chronic disease (hypertension; diabetes) patients due to complaints of long waiting times | Recommendations to employ locum staff with facility-level funds in response to staff shortages | • Reports to CHMT by SCHMT on public queries raised during county budgeting and planning |
| Commodities and infrastructure issues | • Purchase of drugs and supplies with facility-level funds to respond to drug stock-outs | Redistribution of drugs across PHC facilities for drug stock-outs | • Requests to political representatives for capital-intensive infrastructure and equipment related requests by both SCHMTs and HFCs |

Abbreviations: CHMT-County Health Management; HCW-Health Care Worker; HFC-Health Facility Committee; PHC-Primary Health Care; SCHMT-Sub-county Health Management Team

facility and sub-county level, and escalation to higher system levels or to political actors. Many responses were on a case-by-case basis.

Responding to public feedback involved multiple actors across levels and a mix of formal and informal interactions. For example, at the PHC facility level, HFCs commonly delegated to the facility-in-charge the role of dialogue with HCWs when poor HCW conduct was reported to them. This dialogue was usually informal and involved *'sitting down to talk to the colleague'*. If persistent, feedback concerning HCW conduct was handled by the SCHMT and a formal response like a transfer (combined with dialogue with the HCW) was generated. Two case study HFCs (2A and 2B) responded to public feedback related to drug stock-outs and staff shortage by approving the use of facility funds to purchase drugs and hire locum staff, with SCHMT approval. PHC facilities operated bank accounts from where they received funds to support their day-to-day recurrent expenses. SCHMTs were not accounting units and did not operate bank accounts. However, they accessed in-kind resources, for example, during persistent drug stockouts, SCHMT members organised utility vehicles to transport borrowed drugs across facilities in their sub-county. Despite these efforts, drug and staffing gaps were often issues to which the HFCs and SCHMTs could not generate long-term and system-wide responses.

Indeed, few of the responses generated spread across departments or facilities. However, SCHMT-2 highlighted one instance where their response to public feedback resulted in change across all the facilities in their subcounty. This change was in response to public concerns about access to immunisation services on select days of the week in certain PHC facilities. Upon receiving these concerns from the public, SCHMT-2 during their monthly meetings with PHC facility managers recommended that all the facilities in Sub-County-2 should offer immunization services on all the days of the week rather than on select days. This response reportedly enabled the facilities to meet the public's demand for the service as well as to meet facility-level targets for reduction of missed vaccination opportunities. Repeated messaging during the monthly facility-in-charges meeting supported adoption of this practice (immunisation services offered every day) across all the facilities in the sub-county.

**Intangible software and hardware barriers interacted to constrain generation of responses.** Responding to public feedback was constrained by intangible software elements such as communication, power, and provider norms. Information appeared to be controlled by those in more powerful positions; for example, SCHMT members acknowledged that they sometimes attended public participation meetings to represent the CDoH, but they lacked information about the consolidated health sector budget and plan. Thus, they were unable to comprehensively answer questions raised by the public. This was linked to communication challenges that limited how much information was shared with the SCHMT from the CHMT, and a tendency for information to flow upwards (*Quotes 6 and 7, Table 4*).

Failure to feedback information was not only experienced at the County/Sub-county team interface. From interview and FGD data, it was rare for the public to receive communication about actions taken in response to their feedback. Across both SCHMTs and HFCs, there was a perception that responses to public feedback would be apparent when the public saw or experienced a change in service delivery at the facility level. At HFC-level, reluctance to report back to the public concerning actions taken in response to the feedback was underpinned by the uncertainty of change despite promises of action by health managers.

We identified provider professional norms characterised by low information-giving and low receptivity to public feedback. 'Norm' here refers to informal unwritten practices and attitudes (and not documented standards and regulations) held by health providers. Receptivity refers to the willingness to consider or accept feedback from the public [35]. Low information-giving was commonly discussed by SCHMT members in terms of negative communication

experiences where the public received little attention and time. SCHMT-A respondents highlighted how common it was for the public to receive little or no communication about their own or their family member's health status, and situations such as the absence of drugs in facilities.

Low receptivity to public feedback appeared to be linked to 1) a widespread perception among HCWs that the public had a low understanding of health system functioning and 2) health care working conditions. Concerning the former, health managers reported sometimes *'ignoring'* public feedback because it was *'incoherent and they had to balance with existing health system side plans' (County Health Manager)*. County and sub-county health managers noted that frontline providers worked under difficult conditions in which they were under-staffed and where many experienced burn-out. In their view providers' inadequate information-giving and harsh language were linked to the high workload, which limited the time they had to engage with service users (*Quote 8*, *Table 4*). At facility-level, similar issues related to working conditions and public feedback were raised. In facility-1A, the facility-in-charge on receiving public feedback from HFC-1A community members perceived that *'the public complained too much'*, yet the facility in-charge perceived that staff were doing the best they could.

Low receptivity to public feedback and low information-giving reflect provider norms underpinned by two system issues (understaffing (hardware) and hierarchical relationships between the public and health providers (intangible software)) that interacted to perpetuate low responsiveness to public feedback. These provider norms were reportedly deeply ingrained from professional training where little attention was given to how to communicate to the public and patients. Health managers also reported that hierarchical interactions between instructors and students were later replicated in command-and-control interactions between health providers and the public (*Quote 9*, *Table 4*).

**Weakly functioning formal feedback mechanisms contributed to use of informal feedback mechanisms for receiving and responding to public feedback.**   Given the challenges with the functionality of formal feedback mechanisms, the public resorted to voicing their views and concerns through a range of informal feedback mechanisms such as direct phone calls and messages to higher system levels (the CHMT, senior county officials or politicians), social media, and public discussions in social gatherings such as funerals. SCHMT members perceived these informal mechanisms to be disruptive, and that feedback channelled through them was *'often exaggerated and difficult to substantiate'*. Such feedback also generated political pressure (*Quote 10*, *Table 4*), requiring CHMT and SCHMT to quickly investigate and share a report to higher-level actors, usually at the expense of previously planned activities.

The use of informal mechanisms, particularly where political representatives called county health managers to share public concerns, generated some tensions. For example, SCHMT-2 members perceived it to be political interference, reporting that some members of the public used their political connections to be prioritised for services *(Quote 11, Table 4)*. SCHMT-1 members however, also acknowledged that the oversight role of political appointees and elected representatives could be a mechanism to promote better understanding of health system functioning among the public and to generate responses to public feedback. SCHMTs' discomfort with political oversight appeared to stem from the approach used by several MCAs, who either bypassed facility in-charges and sub-county health managers to engage high level officials such as the County Executive Committee member for health (Fig 1) or who showed up unannounced at health facilities to confront staff about public complaints. MCAs on the other hand perceived that facility-in-charges and health managers were often too slow to respond or withheld information when public feedback was unfavourable.

Despite the seemingly antagonistic interactions between MCAs and health managers, some MCAs were able to generate responses to public feedback at facility level. For example, most

MCAs attempted to respond to HFC requests related to supplies, equipment, and infrastructure. However, these responses, especially those that included the purchase of supplies by MCAs, were usually one-off or short-term.

**Responses to vulnerable groups.** Given that the content of feedback rarely included the concerns and views of vulnerable groups, there were few visible responses targeted at vulnerable groups. However, where observed responses to feedback from vulnerable groups appeared to be generated in collaboration with NGOs who worked with these groups. For example, in SCHMT-2, young mothers who raised concerns about inadequate resources for providing care for their infants during their antenatal and post-natal visits were linked to a livelihoods programme run by an NGO that worked in the health system to improve maternal and child health indicators. Similarly, an NGO concerned with the welfare of PLWD had supported the construction of toilets in public health facilities that were disability-friendly, in one of the facilities supported by SCHMT-2. This change however had not spread to all the other primary healthcare facilities within the sub-county.

## Discussion

In this study, we explored responsiveness practice using a systems lens which included: exploring the influence of interacting organisational hardware and software elements, considering feedback interactions between the public (beyond patients) and the health system, and exploring the functioning of formal and informal mechanisms across sub-county (SCHMT) and facility management (HFC) levels. We found multiple feedback mechanisms, but with limited functionality, that commonly functioned in isolation, and inadequately represented vulnerable groups. Our analysis also revealed organisational capacity gaps such as inadequate funding for and staffing of feedback mechanisms (hardware), and absence of clear procedures and guidelines (tangible software). The latter in particular raises questions concerning the extent to which public feedback is valued (intangible software).

Formal feedback mechanisms have received much attention while only a handful of studies have explored informal mechanisms [33, 34, 43, 44]. Our study adds to the literature on informal feedback and illustrates that despite the tensions associated with informal feedback, informal feedback was sometimes useful in supporting service delivery, at least in the short term. We further demonstrate that informal and formal feedback mechanisms are often intertwined in practice. However, informal feedback can become a disincentive for responsiveness, particularly when it results in direct negative repercussions for health providers and managers without accounting for health system weaknesses. Similar findings were reported in a realist review and a study on informal feedback in Malawi that found health providers and managers were more receptive to public feedback in contexts where they felt safe, supported, and their concerns about working conditions heard [33, 35]. Such an environment can be promoted by deliberately planned opportunities for the public (including politicians) to engage with health managers to get a more complete picture of the environment in which health services are generated. Further, clarifying that receiving and responding to public feedback goes beyond fault-finding to include learning opportunities that can shape service delivery could contribute to greater receptiveness among health providers and their managers.

Given the intertwined nature of formal and informal feedback, efforts to strengthen formal mechanisms could consider how to better integrate informal feedback into the health system as suggested by Khan et al. [2]. For example, HFCs could be points at the local level where varied forms of feedback such as satisfaction survey results, direct informal feedback and suggestion box findings are discussed to support facility-wide improvements and change. SCHMTs are also a possible space of integration of concerns and views received through both interpersonal

interactions (informal) and formal mechanisms. To understand how well these mechanisms work together, we suggest further exploration of the linkages between formal and informal feedback mechanisms considering actors, their interactions, motivations, and the strategies of influence they adopt. Such an exploration could help identify how to shift unconstructive interactions between the public and health system actors to be more constructive, and how local alliances can support incorporation of public feedback towards a responsive health system.

Interactions between NGOs and SCHMTs were noted to be important to responsiveness, albeit with mixed effects. These findings are consistent with the literature on tensions between NGO priorities and broader health system goals [45, 46]. First, on one hand, SCHMTs relied on NGO support for functioning of feedback mechanisms (such as HFCs and CHW programmes). Training and availability of stipends contributed to well-functioning HFCs and CHWs who were able to pick up public feedback and transmit to facility-in-charges and SCHMTs. On the other hand, SCHMT members attended many meetings or training sessions planned by NGOs, sometimes leading to lack of quorum for the weekly SCHMT meetings where public feedback could be shared and/or discussed for action. This highlights the need to ensure that NGO priorities do not orient attention away from public feedback. Second, in our study and elsewhere where NGOs have supported functioning of feedback mechanisms, NGO support is often time-limited [47]. Thus, while leveraging the resources and technical capacity of NGOs can strengthen responsiveness, consideration of the limited grant periods which NGOs work within is imperative. To achieve sustainable support for the functioning of feedback mechanisms, and to strengthen responsiveness more broadly, NGOs and health managers need to advocate to policy makers and legislators to set aside and protect resources to support feedback mechanisms when a grant period comes to an end. Overall, across all feedback mechanisms, the emergence of a more responsive health system remains theoretical if actors across facility and sub-national levels do not have the necessary resources (hardware) to enable generation of responses and optimum functioning.

However, we also found that low awareness of policy and guidelines contributed to the ad hoc nature of public feedback management, a finding consistent with literature from other contexts [48–51]. Policy and guidelines can clarify procedures for management of public feedback and enhance organisational commitment to responsiveness by clarifying staff responsibilities at different health system levels, and how feedback can be utilised to improve service delivery [49]. Yet, policies and guidelines by themselves are not sufficient to support adequate functioning; organisational culture and provider norms are important influences on if and how patient and public feedback is responded to [52, 53]. This was illustrated in our study by findings about the provider norms, perceptions, and priorities that oriented health providers and managers away from public feedback. This underscores the importance of going beyond resource allocation to address software dimensions in efforts to strengthen responsiveness.

Our study findings suggest that power, an intangible software element, has significant influence on responsiveness. Power inequities can be reinforced between systems and communities, between health system hierarchies, and within communities [26, 44] particularly undermining equity in responsiveness. The equity dimension of responsiveness requires consideration of which groups provide feedback and whether marginalised or vulnerable groups gave feedback [1, 2]. In our study it was uncommon to 'hear' the voices of vulnerable groups except in cases where parallel or separate feedback channels were set up by NGOs. Subsequently few responses seemed to be targeted to vulnerable groups. Our findings are consistent with studies that report lower inclusion of vulnerable and marginalised groups in feedback channels [54–59]. Across these studies inadequate representation of vulnerable groups was linked to pervasive social inequities which are often sustained by structural forms of power [60, 61]. In-depth exploration of power is warranted given that we have highlighted power

dynamics as a central element of intangible software and inherent in the interactions between actors receiving and responding to public feedback. We have therefore explored the influence of power on responsiveness practices in a companion paper (Kagwanja et al, submitted).

Our study findings also suggest that the processes and interfaces around responsiveness appeared to contribute little towards public trust (intangible software) in the health system given that public feedback was often not considered, and when considered, responses were rarely communicated back to the public. These findings resonate with experiences from other LMICs, for example, Loewensen observes, many health systems in low- and middle-income countries have a poor record in feeding information back to communities [62]. A key implication of weakly responsive health systems in our context and in other LMICs is a lack of trust that could undermine i) the willingness of community members to provide feedback and ii) compliance with health system directives or initiatives. Concerning the former, literature demonstrates that if communities think or do not feel that their input has value, they stop providing input. For example, in South Africa and Kenya, community members have raised questions about the value of their feedback, describing participation as *'spectator politics'* and that it often failed to meet stated goals [58, 63]. Across the practices of receiving, processing, and responding to public feedback, the challenges identified brought into sharper focus the constraints to health system responsiveness already existing during routine times. While in non-crisis times there may not be immediate ramifications to weak responsiveness to public feedback, circumstances like the recent COVID-19 pandemic can catalyse latent distrust built through perceptions of insufficient or absent responsiveness and manifest in public displays of resistance to health system directives.

Overall, we have demonstrated differential participation in feedback mechanisms and weak to middling responsiveness of the HFCs and SCHMTs under study. The findings that the structures for participation and feedback experienced challenges in incorporation of public views into health system decision-making, echo experiences of other LMICs of little empowerment and transformation for marginalised groups as a result of participation [64]. Much of the participation literature has often focused on local action, with little attention to broader governance and political systems. Ideas for how to better incorporate public views into health systems may be informed by understanding how processing spaces and feedback link to power structures and political processes at multiple levels beyond the local [65]. This would require drawing on theories of power, participatory approaches and political science. For example, in proposals of how to improve participation and its outcomes, John Gaventa and others have proposed linking participation and feedback structures to political, social and economic spheres and addressing the agency of the public (and vulnerable groups) to strengthen their engagement in feedback mechanisms while also enhancing the receptivity of institutions to public feedback [64–66].

## Study limitations

One critique of case study research is that it provides little basis for generalisation [36]. This work focused on two SCHMTs and their respective linked HFCs. Thus, the findings cannot be generalised to the population from which the cases are derived -all SCHMTs and HFCs across Kenya- given the complexity and context-specific nature of responsiveness. However, the case study approach supports analytic generalizability, where conclusions about relationships between concepts can be drawn that are transferable to other settings [67, 68]. This study did not include the views of members of the public other than those elected to the HFCs. Given their exposure to the health system during their tenure, HFC members could be considered atypical members of the public. The study therefore did not capture the full range of actors

involved in the responsiveness pathway. However, we set out to understand responsiveness from the health system side, and believe these objectives were addressed even in the absence of data collected directly from the broader public. Further, we acknowledge that there are many issues outside of the health system such as macro-level policy, and broader governance context, that contribute to responsiveness and well-functioning feedback mechanisms and processes are only two components.

## Conclusion

Drawing on our conceptual framework and informed by an understanding of health systems as comprising interacting hardware and software components, we have suggested that a systems view of responsiveness incorporates multiple channels and forms of feedback and requires consideration of the capacity of a health system to support responsiveness to public feedback. Such capacity is dependent on: hardware elements such the funding to establish and sustain functional feedback mechanisms, the human resource to receive, process and act on feedback; tangible software components such as procedures for systematically collecting, and analysing feedback; and intangible software elements such as power, provider norms and communication. Policymakers and health managers seeking to strengthen responsiveness could consider, first, supporting the functioning of feedback mechanisms through allocating adequate funding and human resource. Second, clear procedures and guidelines related to systematic management of public feedback, linked to performance requirements, may strengthen existing responsiveness practice, and generate organisational commitment. Third, there is opportunity to re-imagine functioning of feedback mechanisms, HFCs and SCHMTs. Rather than feedback mechanisms functioning in isolation, a more deliberate effort to integrate varied feedback forms including informal feedback at SCHMT and HFC level could support a more holistic response by the health system. For example, the HFC could proactively review multiple sources of feedback from suggestion boxes, satisfaction records, and informal feedback shared locally and directly with HFC members [69]. SCHMTs could be another point of integration of varied feedback given closer attention to: linkages between HFCs, community health committees, NGOs and other feedback mechanisms; how well vulnerable groups are represented within HFC membership; and to tracking responses to public feedback.

## Supporting information

**S1 Checklist. Inclusivity in global research.**
(DOCX)

**S1 File. Interview guide for SCHMT members and later adapted for CHMT and facility in-charges.**
(DOCX)

**S2 File. Observation checklist for SCHMT meetings and activities.**
(DOCX)

## Acknowledgments

We would like to thank Rebecca Wolfe for her support in developing the figure illustrating the conceptual framework. We are also grateful to our study participants who generously shared their perspectives and views with us.

## Author Contributions

**Conceptualization:** Nancy Kagwanja, Lucy Gilson, Benjamin Tsofa, Jill Olivier, Sassy Molyneux.

**Data curation:** Nancy Kagwanja.

**Formal analysis:** Nancy Kagwanja, Sassy Molyneux.

**Funding acquisition:** Lucy Gilson, Benjamin Tsofa, Jill Olivier, Sassy Molyneux.

**Investigation:** Nancy Kagwanja.

**Methodology:** Nancy Kagwanja, Lucy Gilson, Hassan Leli, Sassy Molyneux.

**Project administration:** Nancy Kagwanja.

**Resources:** Lucy Gilson.

**Supervision:** Lucy Gilson, Benjamin Tsofa.

**Validation:** Sassy Molyneux.

**Writing – original draft:** Nancy Kagwanja.

**Writing – review & editing:** Nancy Kagwanja, Lucy Gilson, Benjamin Tsofa, Jill Olivier, Hassan Leli, Sassy Molyneux.

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
