## [Decision Letter · Decision Letter 0]

24 Jul 2024

PGPH-D-23-02524

Understanding health system responsiveness to public feedback at the sub-national level in Kenya

Dear Dr. Kagwanja,

Thank you for submitting your manuscript to PLOS Global Public Health. After careful consideration, we feel that it has merit but does not fully meet PLOS Global Public Health’s publication criteria as it currently stands. Therefore, we invite you to submit a revised version of the manuscript that addresses the points raised during the review process.

We look forward to receiving your revised manuscript.

Kind regards,

Carl Abelardo T. Antonio

Academic Editor

Journal Requirements:

Additional Editor Comments (if provided):

Please take time to provide a point-by-point response to the reviewer feedback, which concerns both methodological concerns, as well as proper contextualization of the paper content for an international audience.

Reviewers' comments:

Reviewer's Responses to Questions

**Comments to the Author**

1. Does this manuscript meet PLOS Global Public Health’s publication criteria? Is the manuscript technically sound, and do the data support the conclusions? The manuscript must describe methodologically and ethically rigorous research with conclusions that are appropriately drawn based on the data presented.

Reviewer #1: Partly

Reviewer #2: Partly

2. Has the statistical analysis been performed appropriately and rigorously?

Reviewer #1: N/A

Reviewer #2: N/A

3. Have the authors made all data underlying the findings in their manuscript fully available (please refer to the Data Availability Statement at the start of the manuscript PDF file)?

Reviewer #1: No

Reviewer #2: No

4. Is the manuscript presented in an intelligible fashion and written in standard English?

Reviewer #1: Yes

Reviewer #2: Yes

5. Review Comments to the Author

Reviewer #1: Understanding health systems responsiveness to public feedback at the sub-national level in Kenya

PGPH-D-23-02524

General comment: This is an interesting and important topic but needs elaboration.

Specific comments

1. The title can be misleading since the study used a case study in Kilifi County, instead of Kenya. Hence, Kilifi County should be in the title.

2. Abstract contains some local-context terminologies e.g. Health Facility committees and sub-county health management teams, which are hard to follow. The abstract should be rewritten.

3. Materials and methods

- Study setting: The authors should explain the connections between institutions stated in the study, such as Health facilities committed, sub-county health management systems dan county, and members of the county assembly as well as the responsibility between the institutions in the health systems.

- The selection of Kilifi County should be more explained instead of the convenience of the researcher due to existing collaboration. The sampling methods should consider relevance and urgency.

- The author should cite references on the qualitative case study that was used.

- Data collection: the authors should explain the content of guidelines for in-depth interviews, focus group discussion, and observation.

4. Results

- Box 1 needs for elaboration. What are the definitions of participatory, bidirectional, and unidirectional mechanisms?

- Table 3 needs for explanation in the text.

- Fig 3 needs for explanation in the text.

5. Discussion

- Need to add implications for low-and middle-income countries.

- Study limitations of the case study should be elaborated

Reviewer #2: This paper offers an important and necessary exploration of the challenges of health system feedback systems in Kenya. It surfaces really important soft and hard barriers to better feedback systems. The way the study highlights informal and formal mechanisms is important and quite novel.

Most of my concerns with the paper are around the rigor and appropriateness of the theoretical frames and analysis. The primary analysis focuses on domains of organizational capacity, using a software / hardware terminology that is especially confusing. Perhaps instead use intangible / tangible? Or even the use of hard vs. soft capacities would be easier to understand.

*note there is no reference provided for Aragon (which should be listed as Ortiz Aragon?). In addition, I think the Ortiz Aragon piece you are referring to, “Ortiz Aragón, A. A Case for Surfacing Theories of Change for Purposeful Organisational Capacity Development,” actually does not use the terms software or hardware either. In addition, the framework from Aragon that is being used here is not adequately explained in order to allow readers who are not familiar with it to understand your usage.

It seems that perhaps using a power analysis might be a more helpful framework through which to analyze some of these results. It’s mentioned fairly briefly on p. 24 but a more extensive power analysis throughout might be helpful. Simply categorizing power as an “intangible software element” does not do it justice, I think.

Fig 1's conceptual framework does not really include adequate information on how feedback is solicited / offered from patients / marginalized groups. Perhaps consider integrating some of the flows represented in Box 1 into the Fig. 1.

Theoretically, it might also be useful to explore some of the longer histories (and failures) of participation initiatives in African states, such as those by Cooke and Kothcari (2001), which described participatory approaches as a “new tyrrany.” The paper also reminded me of earlier histories of efforts to decentralize and democratize HIV program implementations, such as Nora Kenworthy’s Mistreated (2017) (which discusses at length local institutions very similar to the HCFs). To this end, it might be helpful to think about why similar structures for participation and feedback keep getting (re)created in different waves of global health implementation over decades, but with similarly dismal results when it comes to improving citizen participation in health systems.

While exclusion of marginalized groups is highlighted in the paper abstract and introduction as a major focus of the piece, it receives very limited space in the actual results section (mostly on p. 20-21). I would suggest giving this some greater attention in the discussion of results if you intend to keep it as a major focus of the findings. Additionally, thinking about how power / hierarchy / stigma play here is important.

6. PLOS authors have the option to publish the peer review history of their article (what does this mean?). If published, this will include your full peer review and any attached files.

**Do you want your identity to be public for this peer review?** For information about this choice, including consent withdrawal, please see our Privacy Policy.

Reviewer #1: No

Reviewer #2: No

---

## [Decision Letter · Decision Letter 1]

9 Oct 2024

Understanding health system responsiveness to public feedback at the sub-national level: insights from Kilifi County, Kenya

PGPH-D-23-02524R1

Dear Dr Kagwanja,

We are pleased to inform you that your manuscript 'Understanding health system responsiveness to public feedback at the sub-national level: insights from Kilifi County, Kenya' has been provisionally accepted for publication in PLOS Global Public Health.

Best regards,

Carl Abelardo T. Antonio

Academic Editor

Reviewer Comments (if any, and for reference):

Reviewer's Responses to Questions

**Comments to the Author**

1. If the authors have adequately addressed your comments raised in a previous round of review and you feel that this manuscript is now acceptable for publication, you may indicate that here to bypass the “Comments to the Author” section, enter your conflict of interest statement in the “Confidential to Editor” section, and submit your "Accept" recommendation.

Reviewer #2: All comments have been addressed

2. Does this manuscript meet PLOS Global Public Health’s publication criteria? Is the manuscript technically sound, and do the data support the conclusions? The manuscript must describe methodologically and ethically rigorous research with conclusions that are appropriately drawn based on the data presented.

Reviewer #2: Yes

3. Has the statistical analysis been performed appropriately and rigorously?

Reviewer #2: Yes

4. Have the authors made all data underlying the findings in their manuscript fully available (please refer to the Data Availability Statement at the start of the manuscript PDF file)?

Reviewer #2: (No Response)

5. Is the manuscript presented in an intelligible fashion and written in standard English?

Reviewer #2: (No Response)

6. Review Comments to the Author

Reviewer #2: The authors have more than adequately addressed my comments. I look forward to seeing both this paper and the other one they mention on power analysis in print in the near future.

7. PLOS authors have the option to publish the peer review history of their article (what does this mean?). If published, this will include your full peer review and any attached files.

**Do you want your identity to be public for this peer review?** For information about this choice, including consent withdrawal, please see our Privacy Policy.

Reviewer #2: No
